# DECRYPT trial: study protocol for a phase II randomised controlled trial of cognitive therapy for post-traumatic stress disorder (PTSD) in youth exposed to multiple traumatic stressors

Leila Allen,[1] Polly-Anna Ashford,[2] Ella Beeson,[3] Sarah Byford [iD],[4] Jessica Chow [iD],[1] Tim Dalgleish,[5,6] Andrea Danese,[7,8] Jack Finn,[1] Ben Goodall,[9] Lauren Grainger,[1] Matthew Hammond,[2] Ayla Humphrey,[6] Gerwyn Mahoney-Davies,[10] Nicola Morant [iD],[11] Lee Shepstone,[2] Erika Sims,[2] Patrick Smith [iD],[12] Paul Stallard,[13] Annie Swanepoel,[3] David Trickey,[14] Katie Trigg,[1] Jon Wilson,[15] Richard Meiser-Stedman [iD] [16]

For numbered affiliations see end of article.

**Correspondence to**
Professor Richard Meiser-Stedman;
r.meiser-stedman@uea.ac.uk

## ABSTRACT

**Background** Post-traumatic stress disorder (PTSD) is a distressing and disabling condition that affects significant numbers of children and adolescents. Youth exposed to multiple traumas (eg, abuse, domestic violence) are at particular risk of developing PTSD. Cognitive therapy for PTSD (CT-PTSD), derived from adult work, is a theoretically informed, disorder-specific form of trauma-focused cognitive–behavioural therapy. While efficacious for child and adolescent single-event trauma samples, its effectiveness in routine settings with more complex, multiple trauma-exposed youth has not been established. The Delivery of Cognitive Therapy for Young People after Trauma randomised controlled trial (RCT) examines the effectiveness of CT-PTSD for treating PTSD following multiple trauma exposure in children and young people in comparison with treatment as usual (TAU).

**Methods/design** This protocol describes a two-arm, patient-level, single blind, superiority RCT comparing CT-PTSD (n=60) with TAU (n=60) in children and young people aged 8–17 years with a diagnosis of PTSD following multiple trauma exposure. The primary outcome is PTSD severity assessed using the Children's Revised Impact of Event Scale (8-item version) at post-treatment (ie, approximately 5 months post-randomisation). Secondary outcomes include structured interview assessment for PTSD, complex PTSD symptoms, depression and anxiety, overall functioning and parent-rated mental health. Mid-treatment and 11-month and 29-month post-randomisation assessments will also be completed. Process–outcome evaluation will consider which mechanisms underpin or moderate recovery. Qualitative interviews with the young people, their families and their therapists will be undertaken. Cost-effectiveness of CT-PTSD relative to TAU will be also be assessed.

**Ethics and dissemination** This trial protocol has been approved by a UK Health Research Authority Research Ethics Committee (East of England–Cambridge South, 16/EE/0233). Findings will be disseminated broadly via peer-reviewed empirical journal articles, conference presentations and clinical workshops.

**Trial registration** ISRCTN12077707. Registered 24 October 2016 (http://www.isrctn.com/ISRCTN12077707). Trial recruitment commenced on 1 February 2017. It is anticipated that recruitment will continue until June 2021, with 11-month assessments being concluded in May 2022.

## Strengths and limitations of this study

► This study will involve a highly pragmatic evaluation of a psychological therapy (cognitive therapy for post-traumatic stress disorder (PTSD)) for children and young people with PTSD, being embedded in frontline UK National Health Service child and adolescent mental health services (CAMHS).

► The trial will include youth with the more severe, multiple trauma histories typically seen in CAMHS, and in addition to questionnaire and interview measures of PTSD, the trial will consider complex PTSD symptoms.

► The trial will explore putative mediators and moderators of treatment outcome, and also includes a cost-effectiveness evaluation and qualitative data collection.

► There is no standard 'treatment as usual' for this population, and it is anticipated that treatment in this control arm will vary.

## INTRODUCTION
### Background
Post-traumatic stress disorder (PTSD) is a deeply distressing and disabling psychiatric disorder. PTSD in youth is usually comorbid with other psychiatric conditions,[1 2] and may

persist for years or even decades if untreated.[3 4] The UK Royal College of Psychiatrists estimates[5] the prevalence of PTSD in UK youth to be 3%, while a recent epidemiological study suggested that over 7% of UK youth will have developed PTSD at some point by the age of 18 years.[6]

Traumatic events, defined[7] as those that involve exposure to actual or threatened death, serious injury or sexual violence, are experienced by up to two-thirds of children by age 16 years.[8 9] A significant minority of trauma-exposed children and adolescents go on to develop PTSD.[1 10] For those young people subjected to *multiple* traumatic stressors (eg, physical or sexual abuse, witnessing domestic violence, war or community violence), the clinical presentation of PTSD is particularly severe.[11 12] The life-long impact of multiple trauma exposure in childhood is well documented, with dramatically increased prevalence of a range of emotional disorders, compromised educational, occupational and social functioning, increased use of mental health services and poor physical health.[13–18] PTSD has been shown to have a uniquely harmful role in determining the long-term response to childhood trauma exposure, by mediating the relationship between such experiences and a range of physical and mental health outcomes later in childhood[19 20] and adulthood,[21–23] including psychosis.[24]

In adults, empirical support for trauma-focused cognitive–behavioural therapy (TF-CBT) is substantial.[25–27] TF-CBT typically comprises several core elements: psychoeducation, exposure (to help desensitise patients to trauma memories), cognitive elements (to reframe the meanings and interpretations associated with trauma and its aftermath), and coping management (eg, problem-solving, anxiety management). Over the past 15 years, several randomised controlled trials (RCTs) have been conducted that support the efficacy of TF-CBT (using a variety of specific manuals) in children and adolescents.[28] While this evidence shows considerable promise, a number of important issues remain to be resolved.

First, existing trials rarely compare an experimental treatment with treatment as usual, that is, active treatment/clinical care from a child and adolescent mental health service (CAMHS). Moreover, many studies have not focused on cases of the kind typically referred to mental health services, instead focusing on youth identified in child protection/social services settings. Similarly, many trials have not used 'frontline therapists', reducing the ability to implement trial findings in routine settings.

Second, the cost-effectiveness of implementing a trauma-focused psychological intervention (such as different forms of TF-CBT) in youth mental health services has received little consideration. Evaluating the health economic implications of the investment of time in training and supervising therapists in delivering such specific treatments is essential for the future service delivery.

Third, it is important to consider whether a cognitive therapy for PTSD (CT-PTSD) treatment protocol, a particular form of TF-CBT, can be used in child and adolescent mental health services by frontline therapists. Step increases in treatment efficacy for adults with PTSD have been achieved through careful individual formulation and the enhanced use of theoretically derived techniques in CT-PTSD.[29–31] The cognitive model of PTSD has been largely supported in children and youth.[32] While CT-PTSD adapted for youth[33 34] has been shown to be efficacious for PTSD following single-event trauma when delivered in research clinics,[34 35] its effectiveness in routine clinical settings with youth exposed to multiple traumas has yet to be established. We felt that CT-PTSD may translate well to 'frontline' clinical settings as it employs a formulation-based approach, that is, clinicians are able to tailor session content (ie, particular techniques) to their client's individual presentation which may be particularly helpful when working with clients with more complex histories.

Fourth, the responsiveness of complex PTSD symptoms in children and adolescents to psychological treatment is still poorly understood. Symptoms of complex PTSD include the defining criteria of PTSD alongside disruptions in emotion regulation, relational capacities and self-concept, and are associated with multiple trauma exposures (eg, sexual or physical abuse). While the understanding of complex PTSD in adults has seen significant advances over recent years (with the disorder now included in International Classification of Diseases 11th Revision), little is known about the nature and presentation of complex PTSD in youth, and our treatments for these youth lag behind provision for adults. The ability of youth with high levels of complex PTSD symptoms to derive benefit from psychological therapies for PTSD is largely unknown. However, the first study to address this issue in youth found that complex PTSD symptoms were responsive to TF-CBT, and that the presence of complex PTSD did not lessen the efficacy of TF-CBT to treat PTSD symptoms.[36]

Fifth, moderators and mediators of treatment responsiveness are still poorly understood. While change in negative trauma-related misappraisals has been shown to underpin (ie, mediate) treatment response for CBTs[34 35] (hence our specific secondary hypothesis regarding this mechanism), other potential mechanisms or moderating factors have received scant or no attention. Potential contraindicative moderators for CBTs for PTSD in youth have received comparatively little attention. Shedding light on these factors would inform decision-making about treatments for youth with PTSD and clarify the essential 'ingredients' of successful treatment and barriers to change.

Sixth, the lived experiences of youth undergoing treatment for PTSD following multiple traumas have received little consideration. Qualitative methods may offer important insights into the acceptability and feasibility of delivering treatments like CT-PTSD for youth with PTSD. Such insights may direct how treatments are evaluated and delivered in the future, as well as informing clinician attitudes to the management of this population's needs.

## Objectives

This study aims to fill these gaps in the literature by addressing several objectives. The primary objective of the study is to evaluate whether CT-PTSD is an effective treatment for PTSD symptoms in youth aged 8–17 years old who have been exposed to multiple traumatic stressors, relative to treatment as usual (TAU) in UK National Health Service (NHS) child and adolescent/youth mental health services. The primary outcome is the Child Revised Impact of Event Scale (8-item version; CRIES-8) score at post-treatment. The CRIES-8 is the routine outcome monitoring measure for PTSD in UK CAMHS. We hypothesised that CT-PTSD will be superior to TAU at post-treatment (approximately 5 months post-randomisation) with respect to scores on the CRIES-8. The secondary objectives of the study are as follows: (1) Is CT-PTSD effective in the treatment of complex PTSD, anxiety, depression, general functioning and parent-related mental health in youth with PTSD, relative to TAU?; (2) Is the treatment cost-effective, relative to TAU?; (3) If effective relative to TAU, by what mechanisms does CT-PTSD have its effect?; (4) What demographic, symptomatic, cognitive and psychosocial factors moderate response to treatment?; (5) What are the views and experiences of youth with PTSD, their parents and therapists, about receiving or delivering CT-PTSD, and how do these inform judgements of the acceptability and feasibility of CT-PTSD? In particular, we hypothesised that CT-PTSD would be superior to TAU with respect to complex PTSD symptoms, anxiety, depression, general functioning and parent-related mental health in youth with PTSD; CT-PTSD would be cost-effective relative to TAU; and that the efficacy of CT-PTSD would be mediated through change in trauma-related appraisals.

Given the significant uncertainties over the effect size associated with both CT-PTSD and TAU in this context and the ability of therapists in settings to deliver CT-PTSD with this relatively complex population, this study was considered to be a phase II trial. It is anticipated that the trial will inform the development of a later definitive trial of CT-PTSD for this population.

## Study design

This study is a two-arm, patient-level, single blind, superiority RCT comparing CT-PTSD with TAU. Participants will be allocated to CT-PTSD or TAU according to a 1:1 ratio, with stratification by baseline CRIES-8 score and recruiting site.

This protocol has been written in accordance with the Standard Protocol Items: Recommendations for Interventional Trials 2013 statement.[37]

## METHODS: PARTICIPANTS, INTERVENTIONS AND OUTCOMES
### Study setting

Trial data collection, randomisation, blinding and data analysis will be overseen by the Norwich Clinical Trials Unit (NCTU). Participants will be recruited from NHS CAMHS and youth mental health services in England and Wales.

## Eligibility criteria

A total of 120 children/young people aged 8–17 years will be randomised to either CT-PTSD (n=60) or TAU (n=60). Youth are eligible to be included in the study if they: (1) meet the criteria for a diagnosis of PTSD, as defined by the Diagnostic and Statistical Manual of Mental Disorders, Fifth Edition (DSM-5)[7]; (2) score 17 or greater on the CRIES-8; and (3) have been exposed to multiple traumatic stressors.

Exclusion criteria are: (1) *change* of prescribed psychiatric medication within the past 2 months (though receiving medication was not an exclusion criterion); (2) pervasive developmental disorder or neurodevelopmental disorder (eg, autism, but not attention deficit hyperactivity disorder); (3) intellectual disability; (4) another primary psychiatric diagnosis or clinical need that warrants treatment ahead of PTSD (eg, psychosis, suicidal behaviour, conduct disorder); (5) inability to speak English; (6) ongoing exposure to threat (eg, living with an abuser; regularly placing self in danger) or safeguarding issues; (7) strong likelihood of being unable to complete treatment (eg, imminent house or foster placement move); or (8) history of organic brain damage.

## Interventions

*CT-PTSD.* CT-PTSD for PTSD is a structured, fully manualised[33] psychological treatment delivered in an individual format for children and adolescents. The proposed frequency and duration of treatment is up to 15 treatment sessions (typically 10–12, of 60–90 min duration).

CT-PTSD involves several core elements: psychoeducation, with an emphasis on the role of cognitive processes in the onset and maintenance of PTSD; narrative work and imaginal reliving to help develop a coherent trauma narrative; cognitive restructuring (to reframe the meanings and interpretations associated with trauma and its aftermath), and coping management (eg, addressing maladaptive strategies such as thought suppression, rumination and safety-seeking behaviours). Up to three sessions will be allowed for stabilising other comorbid conditions and difficulties (eg, depression or self-harm).

CT-PTSD will be delivered by NHS CAMHS/youth mental health service therapists who will have completed training in CT-PTSD by a member of the trial team. Trial therapists who deliver CT-PTSD must have an appropriate professional qualification (eg, as a nurse, occupational therapist, clinical psychologist, social worker, psychiatrist or British Association for Behavioural & Cognitive Psychotherapies-registered cognitive–behavioural therapist) and be approved by their local site principal investigator to act as a trial therapist. Individual teams will nominate staff members to complete the CT-PTSD training.

CT-PTSD will be delivered wherever is permitted and feasible for local therapists (eg, in NHS mental health

clinics, local general practitioner (GP) surgeries, at home). Supervision will be provided by a clinical psychologist from the trial team. Following the completion of a course of CT-PTSD, usual NHS care arrangements will apply for participants in this arm.

### TAU

Mental health professionals and others involved in the care of the participants in the TAU arm will be encouraged to provide whatever help they deem necessary, for example, general clinical management, supportive counselling, family therapy, medication. TAU must involve an active treatment, that is, it cannot involve being on a waiting-list. Since TF-CBT has been a recommended treatment for PTSD in youth in the UK since 2005,[25] no therapists in the TAU arm were prevented from delivering this intervention; they were not, however, trained by the research team to deliver the CT-PTSD intervention. Therapist contact in the TAU arm would not be prescribed by trial participation in any way, with one exception: the participants will receive no contact with the trained trial therapists delivering CT-PTSD, and their therapist/clinician will not receive supervision for that case from a trained trial therapist.

### Treatment integrity

Trial therapists will be supervised by members of the trial team while working with trial participants. Trial therapists delivering CT-PTSD will record sessions with participants and rate the use of specific treatment techniques in notes and recording sheets. Trial collaborators will oversee the quality assurance of the training and will monitor and ensure treatment adherence.

### Outcomes

The timing of all measures and assessments is presented in figure 1.

### Primary clinical outcome

The primary outcome measure is PTSD severity using the CRIES-8 score at post-treatment (approximately 5 months post-randomisation). The CRIES-8, a validated self-report questionnaire,[38] is the routine outcome monitoring tool for PTSD in children and adolescents endorsed by the UK Children and Young People's programme–Improving Access to Psychological Therapies. Selecting this measure as our primary outcome is therefore consistent with the pragmatic, 'frontline' focus of the present trial, and will allow clinicians to interpret the trial's findings with regard to their routine outcome measures. The CRIES-8 has the advantage of being used internationally in numerous languages, and being impervious to future changes in diagnostic algorithms, having been in constant use for over 25 years. The CRIES-8 will also be completed at baseline, 2.5 months post-randomisation (mid-treatment), and 11 months and 29 months post-randomisation.

### Secondary clinical outcomes

PTSD diagnosis and symptoms using the Child PTSD Symptom Scale for DSM-5, interviewer version (structured interview, psychometric properties made available by authors),[39] with additional items for measuring dissociation and complex PTSD (that is, items addressing disruptions in emotion regulation, relational capacities and self-concept, that are unique to complex PTSD). The Child and Adolescent Trauma Screen will also be used as a self-report questionnaire to address all DSM-5 PTSD symptoms,[40] with additional items measuring dissociation and complex PTSD. Anxiety and depression will be assessed using the Revised Child Anxiety and Depression Scale.[41] Suicidal ideation will be assessed using five items from the Mood and Feelings Questionnaire.[42] Affect regulation and irritability will be assessed by the Affective Reactivity Index (child and parent/caregiver-report versions).[43] Clinician-rated general functioning will be assessed using the Children's Global Assessment Scale.[44] Parent/caregiver-rated mental health and well-being will be indexed by the Strengths and Difficulties Questionnaire.[45] Parent/caregiver-rated borderline personality traits will be assessed using the McLean Screening Instrument for Borderline Personality Disorder, caregiver version.[46]

In order to ascertain whether parent/caregiver mental health moderates their children's ability to benefit from intervention, and whether parent/caregiver mental health improve with their child receiving intervention, clinical outcomes for parents/caregivers will be assessed. Parent/caregiver depression, anxiety and post-traumatic stress will be assessed using the Patient Health Questionnaire,[47] Generalised Anxiety Disorder Assessment[48] and Post-traumatic Stress Diagnostic Scale, DSM-5 version,[49] respectively.

### Health economic evaluation measures

Resource use will be collected using the Child and Adolescent Service Use Schedule (CA-SUS),[50] modified for a PTSD population, a structured economic interview for youth mental health populations. The CA-SUS will be administered by trial team members at baseline and trained assessors (blinded to allocation) at post-treatment and 11-month follow-up assessments. The interview involves both the young person (if aged 12 years or below) and the young person's primary parent/caregiver. Effectiveness will be measured using the youth version of the EuroQol measure of health-related quality of life (EQ-5D-Y),[51] a preference-based, generic measure capable of generating quality-adjusted life years (QALYs).

### Process measures

Items from an existing interview[52] will be used to assess whether a young person is experiencing hearing voices. Social support will be assessed using the Multidimensional Scale of Perceived Social Support.[53] Several mechanisms proposed by cognitive theories for PTSD will be measured: trauma-related appraisals (the Children's

| TIMEPOINT | STUDY PERIOD | | | | | | |
|---|---|---|---|---|---|---|---|
| | Screening | Baseline | Allocation | Post-allocation | | | |
| | $-t_1$ | $-t_2$ | 0 | Mid [2.5m] | Post [5m] | Follow up [11m] | Follow up [29m] |
| **ENROLMENT:** | | | | | | | |
| *Eligibility screen* | X | | | | | | |
| Informed consent | | X | | | | | |
| PTSD severity (CRIES-8) | | X | | X | X | X | X |
| PTSD diagnosis (CPSS-I-5) | | X | | | X | X | X |
| Allocation | | | X | | | | |
| **INTERVENTIONS:** | | | | | | | |
| CT-PTSD | | | | ←——————————→ | | | |
| TAU | | | | ←——————————→ | | | |
| **ASSESSMENTS:** | | | | | | | |
| Demographics & life history | | X | | | | | |
| PTSD severity, DSM-5 (CATS) | | X | | X | X | X | X |
| Anxiety & depression (RCADS) | | X | | | X | X | X |
| Suicidal ideation (MFQ) | | X | | | X | | |
| Affect regulation (ARI) | | X | | | X | X | |
| Functioning (CGAS) | | X | | | X | X | X |
| Parent/caregiver-report SDQ | | X | | | X | X | X |
| Borderline traits (MSI-BPD-C) | | X | | | X | X | |
| Caregiver depression (PHQ-9) | | X | | | X | X | |
| Caregiver anxiety (GAD-7) | | X | | | X | X | |
| Caregiver post-traumatic stress (PDS-5) | | X | | | X | X | |
| Health & social care resource use (CA-SUS) | | X | | | X | X | |
| Health-related quality of life (EQ-5D-Y) | | X | | | X | X | |
| Voices (interview) | | X | | | X | | |
| Trauma-related appraisals (CPTCI) | | X | | X | X | | |
| Trauma memory quality (TMQQ) | | X | | X | X | | |
| Rumination & self-blame | | X | | X | X | | |
| Safety-seeking behaviours (CSBS) | | X | | X | X | | |
| Social support (MSPSS) | | X | | | X | X | |
| Treatment credibility scale | | | | | X | X | |
| Therapeutic alliance (TASC-r) | | | | | X | X | |
| Adverse events | | X | | | X | X | |

**Figure 1** Standard Protocol Items: Recommendations for Interventional Trials diagram detailing trial activities and measures and their timing. ARI, Affective Reactivity Index; CA-SUS, Child and Adolescent Service Use Schedule; CATS, Child and Adolescent Trauma Screen; CGAS, Children's Global Assessment Scale; CPSS-I-5, Child PTSD Symptom Scale for DSM-5, interviewer version; CPTCI, Children's Post-Traumatic Cognitions Inventory; CRIES-8, Child Revised Impact of Event Scale-8; CSBS, Child Safety Behaviour Scale; CT-PTSD, cognitive therapy for PTSD; DSM-5, Diagnostic and Statistical Manual of Mental Disorders, Fifth Edition; EQ-5D-Y, EuroQol measure of health-related quality of life; GAD-7, Generalised Anxiety Disorder Assessment; MFQ, Mood and Feelings Questionnaire; MSI-BPD-C, McLean Screening Instrument for Borderline Personality Disorder, caregiver version; MSPSS, Multidimensional Scale of Perceived Social Support; PDS-5, Post-traumatic Stress Diagnostic Scale, DSM-5 version; PHQ-9, Patient Health Questionnaire; PTSD, post-traumatic stress disorder; RCADS, Revised Child Anxiety and Depression Scale; SDQ, Strengths and Difficulties Questionnaire; TASC-r, Therapeutic Alliance Scale for Children, revised; TAU, treatment as usual; TMQQ, Trauma Memory Quality Scale,

Post-Traumatic Cognitions Inventory),[54] trauma memory quality (the Trauma Memory Quality Scale),[55] trauma-related rumination and self-blame[56] and safety-seeking behaviours (the Child Safety Behaviour Scale).[57] Treatment credibility will be assessed using four items derived from a previous trial.[30] Therapeutic alliance will be assessed using the Therapeutic Alliance Scale for Children, revised (completed by participant).[58]

### Qualitative data
Semistructured interviews will be conducted to explore the perspectives of youth receiving CT-PTSD, family

members and therapists who deliver CT-PTSD. Interview topic guides will be tailored to each of these perspectives, adapted to be age appropriate for youth respondents, and reviewed by patient and public involvement (PPI) representatives. They will explore experiences and perceived impacts of treatment and its constituent components; the acceptability and feasibility of the treatment received/ delivered; and views of being involved in an RCT and the assessment processes involved. Interviews with youth will be conducted after their post-treatment assessment. In line with sample size guidelines for qualitative interviews,[59] sample sizes of 12–15 for youth and family member interviews are planned, sampled to include youth across the trial age spectrum. A similar sample size is planned for trial therapists.

### Participant timeline

Participants will be assessed five times during the study: baseline, mid-treatment (approximately 2.5 months post-randomisation), post-treatment (approximately 5 months post-randomisation) and at 11-month and 29-month post-randomisation follow-up assessments (approximately 6 and 24 months post-treatment; see figure 1).

### Sample size

This is the first trial to compare CT-PTSD (or any other treatment) to routine care in NHS CAMHS and youth mental health settings for children and young people with PTSD following multiple trauma exposure. While two waiting-list controlled trials have been conducted of the CT-PTSD treatment package in research clinics,[34 35] no UK trials have evaluated the effectiveness of routine care for the treatment of PTSD in young people with multiple trauma exposure, or the effectiveness of CT-PTSD as delivered by therapists working in routine clinical settings. As such, no reliable estimates of effect size are available. An attempt was made to estimate likely effect size based on meta-analyses of existing psychological therapies for PTSD in children and adolescents. The waiting-list controlled effect size of CT-PTSD was greater than 1.2 in each of the studies to have examined this intervention to date.[34 35] The Cochrane meta-analysis of treatments for PTSD in children and adolescents[10] yielded an effect size of 1.05 for all psychological therapies versus control conditions, whereas for CBT versus control conditions, the effect size was 1.34. However, these effect sizes are likely to overestimate the efficacy of CBT; few studies were included in the headline analyses, and those that were included were waiting-list controlled, rather than using an active control condition (eg, TAU or other interventions that involved clinician/therapist contact). A high-quality trial of another psychological therapy (prolonged exposure, a form of TF-CBT) for adolescent girls with PTSD following sexual abuse, that used supportive counselling as a control intervention, yielded a controlled (ie, between groups) effect size of 1.0.[39] An alternative meta-analysis,[60] including a broader range of studies to evaluate CBT in youth with PTSD (many of which used

active control treatments such as supportive counselling) suggested a controlled effect size of around 0.67; a more recent meta-analysis[61] suggested an effect size for TF-CBT of 0.79. The more conservative effect size estimate of 0.67 was used for the proposed trial.

In order to have 90% power to detect a between groups effect size of 0.67 (two-tailed t-test, 0.05 significance level), a sample size of 96 (48 participants per group) is required; a combined sample size of 96 would have 80% power to detect an effect size of 0.58 (two-tailed t-test, 0.05 significance level). In order to account for drop-out (estimated at 20%), 120 participants will be recruited.

### Recruitment

Participants will be recruited from NHS Trusts in England and Wales. Clinicians in participating teams will be encouraged to contact the trial team to discuss potential participants to ensure that they meet inclusion criteria. If a child or young person appears eligible, the clinician will share information with the participant and their family about the study who will be given at least 48 hours to decide whether they wish to participate. Consenting families will be contacted by the trial team to discuss further and to arrange baseline assessments; trial eligibility will be confirmed following this assessment. Children and young people will not be able to enter DECRYPT (Delivery of Cognitive Therapy for Young People after Trauma) unless a therapist within the team is available to offer CT-PTSD and another clinician/therapist is available to offer TAU, that is, TAU cannot involve a waiting-list.

## METHODS: ASSIGNMENT OF INTERVENTIONS
### Allocation

Following pretrial assessments, consenting participants will be randomised to study arms stratified by CRIES-8 score (17–28, 29–40) and site (ie, NHS Trust). This cut-off for stratification was based on data from an unpublished pilot study that preceded DECRYPT, where the mean score for study participants was 28.75. An online randomisation service managed by NCTU will assign allocation to groups. Allocation is by preset lists of permuted blocks with randomly distributed block sizes (determined by the trial statistician). The lists will be generated by the Data Management Team at NCTU. The trial manager will enrol participants. Following randomisation, participants, their clinical team and their GP will be notified of their allocation by the trial manager. See figure 2 below for the Consolidated Standards of Reporting Trials diagram.

### Blinding

Trained assessors collecting post-treatment and follow-up data will be blinded to group allocation. These assessments will be undertaken by trained assessors with no other role in the trial. Following allocation to CT-PTSD or TAU, all participants in the study, their care coordinator/referrer and clinical team (if applicable) are asked

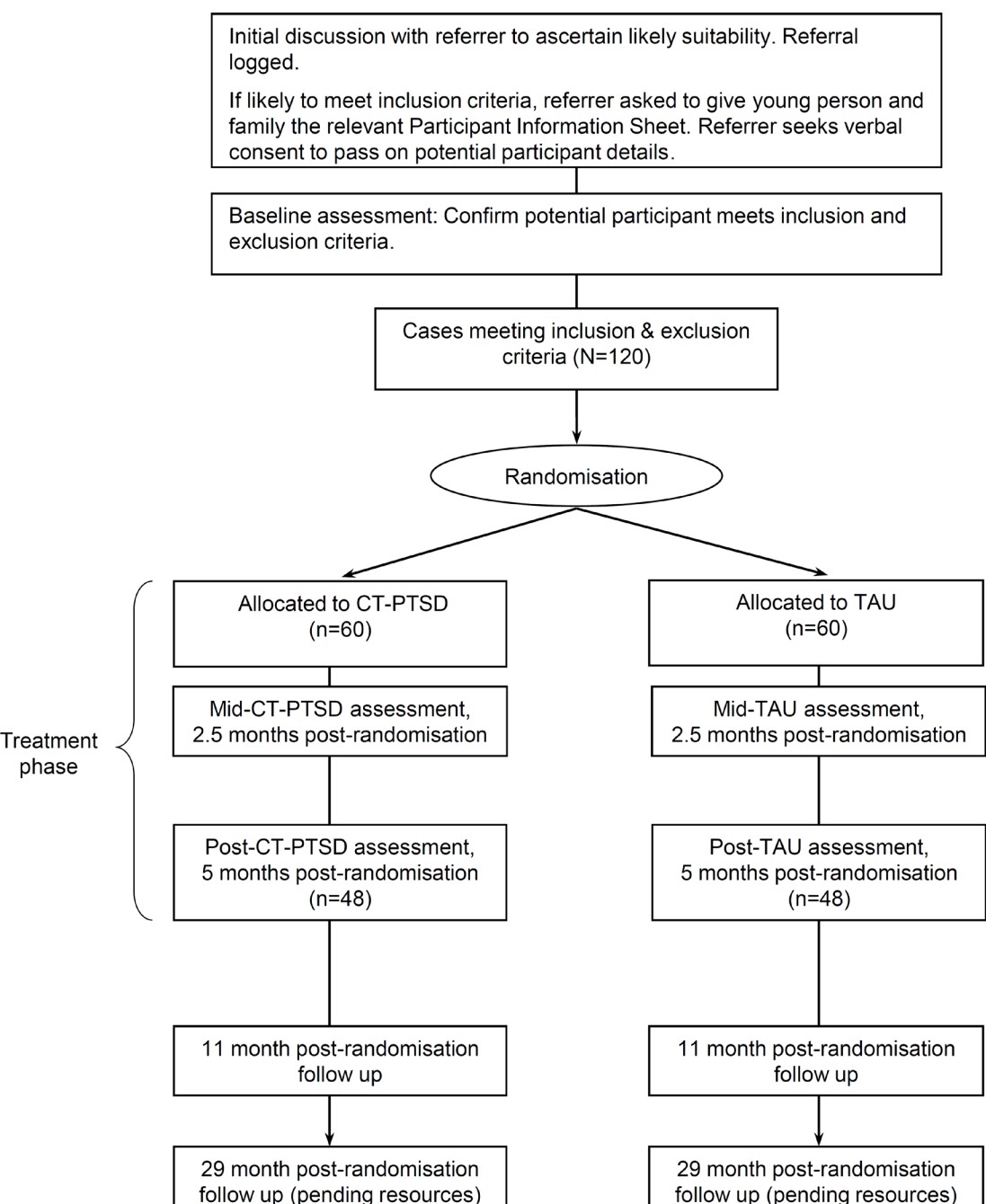

**Figure 2** Trial Consolidated Standards of Reporting Trials diagram. CT-PTSD, cognitive therapy for post-traumatic stress disorder; TAU, treatment as usual.

not to reveal the group to which the participants were randomised to the trained assessor.

## METHODS: DATA COLLECTION, MANAGEMENT AND ANALYSIS
### Data collection methods
Quantitative outcome variable data will be collected using paper forms and entered electronically on to the trial database or online. Qualitative data will be collected using face-to-face interviews, or where this is not possible telephone or video-call interviews. An appropriately trained individual will undertake semistructured interviews with participants and their families after treatment has been

completed. For participants who have withdrawn from treatment or the trial but consent to taking part in a qualitative interview, this will be completed when appropriate. Interviews will be audio-recorded and transcribed verbatim, with all identifying information removed.

### Data management
To maintain confidentiality, each participant will be given a unique trial Participant Identification Number (PIN). Data will be entered under this identification number onto the central database stored on the servers based at NCTU. The database and coding frames have been developed by the Clinical Trial Manager in conjunction

with NCTU. The database software provides a number of features to help maintain data quality, including maintaining an audit trail, allowing custom validations on all data, allowing users to raise data query requests and search facilities to identify validation failure/missing data. After completion of the trial, the database will be retained on the servers of NCTU for ongoing analysis of secondary outcomes.

## Statistical methods

A full statistical analysis plan will be written and agreed with the Data Monitoring Committee (DMC) and Trial Steering Committee (TSC) prior to database lock and any data analysis. This plan will be amended only with agreement of the former two committees.

Our primary analysis will compare CT-PTSD with TAU on CRIES-8 scores at post-treatment. The primary analysis will be on the intention-to-treat principle: that is, all participants will be followed up for data collection irrespective of adherence to treatment and will be analysed according to group allocation rather than intervention received. Assuming a normal distribution (potentially of transformed values), a linear model will be constructed. This will include recruiting site (as a random factor), CRIES-8 at baseline (as a covariate) and any factors considered prognostic and determined in advance of any analysis, together with treatment arm as a fixed effect. Analysis of other secondary outcomes, and of putative moderators and mediators, will be considered exploratory.

The primary intention-to-treat analysis is intended to provide inferences regarding the effectiveness of the intervention overall not to provide inferences regarding the causal effect of the intervention itself, but on the intervention as deployed in 'real life'. Statistical significance will be set at the conventional (two-tailed) 5% level and all parameter estimates will be presented with 95% CIs. Analyses will be carried out by the trial statistician blinded to group identity, (ie, 'subgroup' blind). There are no plans for interim efficacy or subgroup analyses. Analyses will be carried out in SAS (currently V.9.4).

Mediation and moderation analyses will be undertaken following the procedures outlined by Kraemer and colleagues.[62] As with all our secondary analyses, these will be considered exploratory.

## Qualitative analysis

Qualitative data will be analysed using thematic analysis within NVivo software.[63] A collaborative approach throughout the analytical process in which emerging themes are discussed between those who collect data and other members of the research team will enhance reflexivity and validity.[64] The views and experiences of each group (children/young people, parents/caregivers, trial therapists) will be analysed both separately and in comparison with each other in order to gain a multiperspective view of the acceptability and feasibility of delivering CT-PTSD in routine NHS child and youth mental health services. Both commonalities and variations within and between these stakeholder groups will be explored.

## Health economic evaluation

Economic evaluation will compare CT-PTSD with TAU at the 11-month follow-up. It will take the NHS/personal social services perspective, preferred by the National Institute for Health and Clinical Excellence (NICE),[65] but will additionally include education services, given the age of the population. Service use data will be collected using the CA-SUS while health-related quality of life will be assessed using the EQ-5D-Y (see above). Data on CT-PTSD contacts and TAU (CAMHS) contacts and on indirect time for the intervention (eg, supervision, training) will be collected directly from therapists/clinicians and service records; participants will be asked not to mention any CAMHS/mental health contacts so as to reduce the possibility of unblinding and double-counting of service contacts. Service use estimates will be combined with standard UK sources for unit costs to estimate total costs (including NHS reference costs for hospital contacts, the British National Formulary for medications and the Personal Social Services Research Unit Costs of Health and Social Care). The cost of CT-PTSD will be directly calculated.[66]

Economic analysis will be carried out on an intention-to-treat basis using an analysis plan to be drawn up prior to data analysis. Costs and outcomes will be compared and presented in terms of mean differences and 95% CIs obtained by non-parametric bootstrap regression to account for the non-normal distribution commonly found in economic data.[67] Cost-effectiveness will be assessed through the calculation of incremental cost-effectiveness ratios (ICERs) (the additional cost of one intervention compared with another divided by the additional effects). The primary analysis will focus on effects measured using the CRIES-8, known to be sensitive to change in this population. Secondary analyses will explore cost-effectiveness in terms of QALYs derived from the EQ-5D-Y and using the area under the curve approach,[68] a more policy relevant measure that is preferred by NICE, but with unknown sensitivity in young people with PTSD. Uncertainty will be explored using cost-effectiveness planes and cost-effectiveness acceptability curves based on the net-benefit approach.[69 70] These curves are an alternative to CIs around ICERs and show the probability that one intervention is cost-effective compared with the other, for a range of values that a decision-maker would be willing to pay for an additional unit of an outcome. All economic analyses will be adjusted for baseline CRIES-8 score and site, in line with the clinical analyses, plus baseline values of the variables of interest (cost, QALYs), to provide a more relevant treatment effect estimate.[71]

## METHODS: MONITORING AND DATA MANAGEMENT
### Data monitoring

An independent TSC will be responsible for oversight of the trial in order to safeguard the interests of trial

participants. The TSC will provide advice to the chief investigator, NCTU and the trial sponsor (University of East Anglia). A separate DMC comprising three independent researchers (one statistician) will monitor adverse event and adverse reactions. They will undertake a mid-recruitment review of adverse event and adverse reaction data. Their report will be reviewed by the TSC.

### Adverse events

Adverse events refer to unwanted medical events (for example, worsening symptoms) occurring throughout the trial, regardless of whether they are causally related to the trial procedures. For the purposes of this trial, the following would be considered adverse events: increase in extent of self-harm or suicidal ideation; significant worsening in symptoms (ie, an increase in CRIES-8 score equal to or greater than 7); and/or concern regarding decline in mental state. Precautions have been taken to reduce the likelihood of such events occurring; for example, the therapists delivering the CT-PTSD sessions will be trained in how to manage any distress that arises. Therapists in both arms of treatment will have experience in working with complex populations and management of risk. A Safety Management Plan detailing procedures for dealing with adverse events and adverse reactions has been agreed with NCTU.

### Patient and public involvement

The original trial design was formulated with input from several PPI groups. Service users commenting on the PYCES trial (National Institute for Health Research RfPB-funded RCT addressing PTSD in preschool children) also stressed the need to investigate youth with PTSD following multiple trauma exposure. Patient/participant feedback was an essential component of a case series of the treatment protocol (CT-PTSD) for PTSD in this group. Families participating in this study provided extensive feedback on the acceptability of the treatment protocol and the research procedures developed for use with this multiple trauma population. These responses were collated and were used to guide the development of this protocol.

The youth panel of *inspire* (a PPI group hosted by Norfolk and Suffolk NHS Foundation Trust) identified several agencies to involve in the service study and dissemination, and raised questions over the nature of 'TAU' for this population and the adequate 'dosage' of CT-PTSD. The *inspire* youth panel were keen to be represented on the Trial Management Group (TMG) and TSC. The panel were positive about the long-term benefits that may arise from this treatment, the CAMHS setting and the close involvement of PPI groups.

RM-S met with the UK Clinical Research Network Young Person's Mental Health Advisory Group to receive their views on the initial proposed trial design. They raised issues about ongoing post-trial care for trial participants, and the consenting procedure (addressed in the Ethics and dissemination section of this proposal). They

were also concerned about psychotherapy being delivered poorly or in an overly generic way. Group members commented that despite sometimes being disclosed to mental health teams, traumatic experiences were often not addressed in treatment or even mentioned again.

PPI representatives, recruited from the *inspire* group, contributed to the development of the study participant information sheets and consent forms. PPI representatives will be recruited to the TMG, will meet with the trial team on a regular basis to provide more specific input on study procedures and will assist with writing the final report. Trial results will be communicated to all participants through a newsletter.

## ETHICS AND DISSEMINATION
### Research ethics approval and protocol amendments

DECRYPT was approved by a UK Health Research Authority Research Ethics Committee (REC; East of England–Cambridge South, 16/EE/0233). The study personnel, a TMG, a TSC and a DMC, have been established and will ensure that the study is conducted within appropriate NHS and professional ethical guidelines. Good Clinical Practice training will have been undertaken by all those directly involved in running the study. Protocol amendments will require approval from the REC, and where relevant will be passed on to the trial register (ie, ISRCTN). To date, protocol amendments have been made to: clarify CT-PTSD therapist background requirements, include screening of electronic notes in one NHS Trust, share core baseline assessment data with clinical teams in both arms of the trial (to avoid unnecessary duplication of assessments), provide clarification on where identifiable participant data would be stored, and clarify what would constitute an expected event rather than an adverse event or adverse reaction (major amendment, March 2017); removal of some secondary process outcome measures that felt to be overly burdensome for participants (intelligence testing, emotional regulation strategies); clarifying that participants would receive payment for completing qualitative interviews in addition to the main trial assessments (major amendment, January 2018); and other procedural changes, such as trial team changes and the additional of students to the trial team. In light of the COVID-19 situation, an amendment was made to allow electronic (ie, remote) consenting.

### Consent and assent

For youth aged under 16 years, informed consent will be provided by parents and caregivers, and the child or young person will also be asked to give their assent for trial entry. Youth aged 16 years or older can provide informed consent without their parent or caregiver's involvement. Additional consent is required for the audio-recording of therapy sessions. Additional consent forms will also be used for qualitative data collection, to ensure that participants agree to the interviews being recorded and

their anonymised data (ie, quotes) being used in future publications.

## Confidentiality

The trial database will be password protected and only accessible to members of the DECRYPT trial team at NCTU, and external regulators if requested. The servers are protected by firewalls and are patched and maintained according to best practice. The physical location of the servers is protected by CCTV and security door access. Data will be entered in the approved DECRYPT database by a member of the DECRYPT trial team at NCTU or by assessors at each site, and protected using established NCTU procedures.

The identification, screening and enrolment logs, linking participant identifiable data to the pseudoanonymised PIN, will be held locally by the trial team. This will either be held in written form in a locked filing cabinet or electronically in password-protected form on secure computers. After completion of the trial, the identification, screening and enrolment logs will be stored securely by the sites for 10 years unless otherwise advised by NCTU.

## Declaration of interests

Some investigators in DECRYPT provide training in the delivery of CT-PTSD, for which they receive payment (eg, to clinical psychology or CBT training courses, conference workshops).

## Dissemination policy

There are no publication restrictions and findings will be disseminated broadly to participants, healthcare professionals, the public and other relevant groups. The study findings will be published in peer-reviewed journals. Clinical workshops will be offered to practitioners and team leads to share study findings and consider how practice can be improved. The full trial protocol is available from RM-S. Outcomes from the post-treatment and 11-month post-randomisation assessments may be analysed and published prior to the completion of the 29-month assessments.

## DISCUSSION

PTSD in children and adolescents represents a significant public health burden. It is hoped that the DECRYPT trial will provide insight into how some of the most severe PTSD cases in youth might be better treated in a real-world setting, and provide a platform to future pragmatic research in this area.

## Author affiliations

[1] Department of Clinical Psychology and Psychological Therapies, Norwich Medical School, University of East Anglia, Norwich, UK
[2] Norwich Clinical Trials Unit, Norwich Medical School, University of East Anglia, Norwich, UK
[3] Hertfordshire Partnership University NHS Foundation Trust, Hatfield, UK
[4] King's Health Economics, King's College London, London, UK
[5] MRC Cognition and Brain Sciences Unit, Cambridge, UK
[6] Cambridgeshire and Peterborough NHS Foundation Trust, Cambridge, UK
[7] Department of Child and Adolescent Psychiatry, King's College London Institute of Psychiatry, Psychology and Neuroscience, London, UK
[8] Social, Genetic and Developmental Psychiatry Centre, King's College London, London, UK
[9] North East London NHS Foundation Trust, Rainham, UK
[10] Cardiff and Vale University Health Board, Cardiff, UK
[11] Division of Psychiatry, Faculty of Brain Sciences, University College London, London, UK
[12] Department of Psychology, King's College London Institute of Psychiatry, Psychology and Neuroscience, London, UK
[13] Department for Health, University of Bath, Bath, UK
[14] Specialist Trauma and Maltreatment Service, Anna Freud National Centre for Children and Families, London, UK
[15] Norfolk and Suffolk NHS Foundation Trust, Norwich, UK
[16] Department of Clinical Psychology and Psychological Therapies, University of East Anglia, Norwich, UK

**Acknowledgements** The authors gratefully acknowledge the support of the Trial Steering Committee (Sam Cartwright-Hatton, Peter Fonagy,Shirley Reynolds and William Yule) and Data Monitoring Committee members (Barney Dunn, Ian Goodyer and Victoria Vickerstaff). We are very grateful for the important contribution of all the young people and carers who helped to shape this study as it was being designed.

**Contributors** RM-S, LA, TD, PSm, NM, PSt, AD, JW, LS and SB each contributed to the design of the trial. RM-S, LA, LG, KT, JC, PSm, NM, PSt, EB, AD, AH, JW, BG, GM-D, MH, P-AA, DT, JF, AS and ES oversaw recruitment and data collection. LA, RM-S, ES and MH drafted the protocol. All authors read and approved the final manuscript. All authors have agreed both to be personally accountable for the author's own contributions and to ensure that questions related to the accuracy or integrity of any part of the work, even ones in which the author was not personally involved, are appropriately investigated, resolved, and the resolution documented in the literature.

**Funding** DECRYPT is funded by a National Institute for Health Research (NIHR) Career Development Fellowship to RM-S (CDF-2015-08-073). This publication presents independent research funded by the NIHR.

**Disclaimer** The views expressed are those of the author(s) and not necessarily those of the NHS, the NIHR or the Department of Health and Social Care. The NIHR had no role in the design of the study, or the collection, analysis, and interpretation of data, or in writing the manuscript.

**Competing interests** PS authored a treatment manual that is being used in this trial.

**Patient and public involvement** Patients and/or the public were involved in the design, or conduct, or reporting, or dissemination plans of this research. Refer to the Methods section for further details.

**Patient consent for publication** Not required.

**Provenance and peer review** Not commissioned; externally peer reviewed.

**ORCID iDs**
Sarah Byford http://orcid.org/0000-0001-7084-1495
Jessica Chow http://orcid.org/0000-0002-0139-449X
Nicola Morant http://orcid.org/0000-0003-4022-8133
Patrick Smith http://orcid.org/0000-0002-0743-7972
Richard Meiser-Stedman http://orcid.org/0000-0002-0262-623X

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
