## [Reviewer comments · BMJ Open]

ARTICLE DETAILS

TITLE (PROVISIONAL)	The DECRYPT trial: Study protocol for a phase II randomised controlled trial of cognitive therapy for post-traumatic stress disorder (PTSD) in youth exposed to multiple traumatic stressors.
AUTHORS	Allen, Leila; Ashford, Polly-Anna; Beeson, Ella; Byford, Sarah; Chow, Jessica; Dalgleish, Tim; Danese, Andrea; Finn, Jack; Goodall, Ben; Grainger, Lauren; Hammond, Matthew; Humphrey, Ayla; Mahoney-Davies, Gerwyn; Morant, Nicola; Shepstone, Lee; Sims, Erika; Smith, Patrick; Stallard, Paul; Swanepoel, Annie; Trickey, David; Trigg, Katie; Wilson, Jon; Meiser-Stedman, Richard

VERSION 1 – REVIEW

REVIEWER	Samuli Kangaslampi Tampere University, Faculty of Social Sciences / Psychology
REVIEW RETURNED	18-Jan-2021

GENERAL COMMENTS	This study protocol describes a randomized controlled trial on cognitive therapy for PTSD for youth exposed to multiple traumatic events. This research is both clinically and scientifically highly relevant, especially as the study is to be conducted in a pragmatic way in a usual-care environment. Overall, the protocol is thoroughly but concisely presented, with appropriate supporting documentation and checklists. The study appears to have been carefully planned and thought out. As the details of the implementation of the study have already been established and are unlikely to be changed at this stage, my review focuses on the presentation of the protocol, the rationale for some of the choices made, and other similar issues. I have only minor suggestions and questions. 1. As the study has already commenced in 2017, and Page 6 notes that this version of the protocol is from April 2020, changes to the protocol may have happened while the study was on-going. If this is the case, it would be best to list the ways in which the protocol has been amended during implementation and how this might affect its results or validity.2. Outcomes of the trial are to be assessed with a wide range of convincing measures. Perhaps the only surprising choice is to use scores on the CRIES-8, a brief self-report measure that excludes some areas of PTSD symptoms, as the only primary outcome. As the protocol also includes other more comprehensive measures of PTSD symptoms, could the authors provide an explicit rationale for making scores on CRIES-8 the primary outcome?3. The ISRCTN registration of the study includes one preregistered
---

	hypothesis. It would appear reasonable to also present this preregistered hypothesis in this published study protocol. It does not appear that other hypotheses have been preregistered prior to the commencement of the study, e.g., in relation to possible mediating factors, or the secondary outcomes. It would be best to state this explicitly, as well. 4. The section on mediators and moderators on Page 12 could be clearer in separating mechanisms (typically studied as mediators) and moderators of treatment and the importance of studying both. In the objectives, these two points of view are clearly presented, but in this section in its present form, the reader may end up with the idea that “potential contraindications” refers to mechanisms. 5. “there are no plans for formal interim efficacy or subgroup analyses” (p. 22) – This could be read to suggest that interim analyses might be carried out informally. This is likely not the intention. If, as expected, data will be not be looked or analysed during the trial at all, it would be best to state this more clearly. 6. The protocol notes that a full statistical analysis plan will be written separately. However, as there will be many outcomes and many analyses, explicitly noting at least the method that will be used to account for the problem of multiple comparisons in the study protocol itself would be valuable. Considering the increasing emphasis on preregistered analyses, it would also be valuable to briefly state which analyses or hypotheses to be examined later have been preregistered and which will be exploratory by nature.
--	---

REVIEWER	J Rossouw Stellenbosch University Faculty of Medicine and Health Sciences, Psychiatry
REVIEW RETURNED	24-Jan-2021

GENERAL COMMENTS	Congratulations on a very well written study protocol describing a very important study of PTSD treatment amongst adolescents and children. This study will be especially important as the treatment providers will be functioning within their existing roles and not within a specific research setting. Additionally looking at the treatment of children with PTSD after exposure to multiple trauma events as opposed to single-incident trauma exposure will provide needed further evidence, if existing TF-CBT treatments will suffice to assist this group. The inclusion of qualitative data on stakeholders experiences of treatment will add a more individualised and richer telling of the impact of this treatment study on all concerned. The inclusion of measures to determine cost-effectiveness of CT-PTSD relative to TAU is a further strength of this study. I only have 4 areas that require clarification. The first, is that the title and the primary aim of the study is described as to determine if cognitive therapy is effective in the treatment of PTSD of children and adolescents who experienced multiple trauma events. A reference too complex PTSD under unresolved issues in the Background section (p11, line 47) seems to indicate that the study is trying to address both the treatment of children with multiple trauma exposures with a diagnosis of PTSD and also in the process evaluate the characteristics associated with a complex PTSD diagnosis as a secondary aim, such as "disruptions in emotion regulation, relational capacities and self-
---

	concept". There seems to however not be any assessment tool to specifically identify complex PTSD as a diagnosis. References to "additional items for measuring dissociation and complex PTSD" is made. So it seems as if the authors expect to treat both PTSD and complex PTSD by identifying children who experienced multiple stressors who meet the criteria for PTSD. It seems that if one argues for two distinct diagnosis (PTSD and Complex PTSD), as some now do, then one would have to assess the effectiveness of both these conditions in two treatment arms, rather than in a combined treatment arm (PTSD)? Perhaps the authors can clarify their thinking around these related diagnosis and the inclusion criteria only being a PTSD diagnosis. The attached NCTU protocol (2017) also does not have treatment of Complex PTSD as an objective, whereas this manuscript has "CT-PTSD effective in the treatment of complex PTSD" as an objective. This is understandable as the NCTU protocol is dated 2017. This manuscript as an updated version would need to now accommodate the changes around Complex PTSD as a ICD-11 diagnosis (2019) more clearly or at least place this change more clearly in context in terms of the objectives of this study. Secondly, a bit more detail on the planned analysis of secondary measures and mediator and moderator analysis would be welcomed. Thirdly, as the journal require a start date and proposed completion date of active treatment, perhaps the authors can add this to the manuscript. Fourthly, the authors make no direct reference to the most studied TF-CBT intervention in children and adolescents (Cohen, Mannarino and Deblinger) of the same name. An article published in 2012 in Child Abuse and Neglect by Cohen et al. already describe the relevance of this treatment in the management of complex trauma. More recent RCT's involving TF-CBT have mostly included children with multiple traumas (average of 3.4); (Mannarino quoted in an presentation at NCTSN). To date 21 RCT's of this intervention have been completed in multiple settings ranging from LMICS to EU settings (Netherlands, Norway, Germany). Therefore the motivation behind investigating this particular CT-PTSD treatment manual can be motivated in more detail and perhaps a reference to the results of the pilot study that preceded this study could help motivate this choice?
--	---

REVIEWER	Cedric Sachser University of Ulm, Child and Adolescent Psychiatry/Psychotherapy
REVIEW RETURNED	22-Mar-2021

GENERAL COMMENTS	I congratulate the authors on planning such a great trial with real added value for multiple traumatized children and adolescents within the NHS mental health services. The trial protocol clearly describes the roles and responsibilities, the objective and the trial design, the population of interest, interventions, inclusion/exclusion criteria, recruitment process, outcomes, power calculation, assignment of interventions, blinding, data-management, and statistical methods and analyses. Including the attached full trial protocol all relevant information according to the SPIRIT checklist were given.
--

	The authors have registered the study before entry of the first participant. However, I was wondering why the study protocol was not published earlier (recruitment is anticipated to end soon) as a detailed publication of study protocols can reduce publication bias and improve the reproducibility of research. I have only minor comments/questions:  1. How were therapists selected? Were therapists randomized to either be trained in CT-PTSD or not? This may influence the comparison as therapists open the to be trained in CT-PTSD might constitute a “special” group of NHS therapists especially interested in EBTs for PTSD in children and adolescents. 2. How was the stratification value (17-28, 29-40) of the CRIES-8 determined? 3. The authors could reflect a little more on why they have chosen the CRIES-8 as a primary outcome over the CPSS-I-5 severity score.
--	--

VERSION 1 – AUTHOR RESPONSE

Reviewer: 1

As the details of the implementation of the study have already been established and are unlikely to be changed at this stage, my review focuses on the presentation of the protocol, the rationale for some of the choices made, and other similar issues. I have only minor suggestions and questions.

1. As the study has already commenced in 2017, and Page 6 notes that this version of the protocol is from April 2020, changes to the protocol may have happened while the study was on-going. If this is the case, it would be best to list the ways in which the protocol has been amended during implementation and how this might affect its results or validity.

RESPONSE: Thank you for highlighting this issue. We have now added the following text to outline changes made to the protocol (pp23-24).

“To date, protocol amendments have been made to: clarify CT-PTSD therapist background requirements, include screening of electronic notes in one NHS Trust, share core baseline assessment data with clinical teams in both arms of the trial (to avoid unnecessary duplication of assessments), provide clarification on where identifiable participant data would be stored, and clarify what would constitute an expected event rather than an adverse event or adverse reaction (major amendment, March 2017); removal of some secondary process outcome measures that felt to be overly burdensome for participants (intelligence testing, emotional regulation strategies); clarifying that participants would receive payment for completing qualitative interviews in addition to the main trial assessments (major amendment, January 2018); and other procedural changes, such as trial team changes and the additional of students to the trial team. In light of the COVID-19 situation, an amendment was made to allow electronic (i.e. remote) consenting.”

2. Outcomes of the trial are to be assessed with a wide range of convincing measures. Perhaps the only surprising choice is to use scores on the CRIES-8, a brief self-report measure that excludes some areas of PTSD symptoms, as the only primary outcome. As the protocol also includes other more comprehensive measures of PTSD symptoms, could the authors provide an explicit rationale for making scores on CRIES-8 the primary outcome?

RESPONSE: The choice of the CRIES-8 reflects the pragmatic nature of the DECRYPT trial. We have added the following explanation (p13):

“Selecting this measure as our primary outcome is therefore consistent with the pragmatic, “front-line” focus of the present trial, and will allow clinicians to interpret the trial’s findings with regards to their routine outcome measures. The CRIES-8 has the advantage of being used internationally in numerous languages, and being impervious to future changes in diagnostic algorithms, having been in constant use for over 25 years.”

3. The ISRCTN registration of the study includes one preregistered hypothesis. It would appear reasonable to also present this preregistered hypothesis in this published study protocol. It does not appear that other hypotheses have been preregistered prior to the commencement of the study, e.g., in relation to possible mediating factors, or the secondary outcomes. It would be best to state this explicitly, as well.

RESPONSE: Apologies that we did not make this more explicit; we had nominated our primary outcome as the CRIES-8 and the main outcome point as being post-treatment (approximately 5 months post-randomisation) in the Methods-Primary Clinical outcome sub-section. In order to be clearer, we have made this and our overall hypothesis clearer in the Objectives sub-section of the Introduction, to be consistent with the SPIRIT 2013 checklist:

Re: primary outcome, p9:

“The primary outcome is the Child Revised Impact of Event Scale (8-item version; CRIES-8) score at post-treatment. The CRIES-8 is the routine outcome monitoring measure for PTSD in UK child and adolescent mental health services (CAMHS). We hypothesised that CT-PTSD will be superior to TAU at post-treatment (approximately five months post-randomisation) with respect to scores on the CRIES-8.”

Re: secondary outcomes, p10:

“In particular, we hypothesised that CT-PTSD would be superior to TAU with respect to complex PTSD symptoms, anxiety, depression, general functioning and parent-related mental health in youth with PTSD; CT-PTSD would be cost-effective relative to TAU; and that the efficacy of CT-PTSD would be mediated through change in trauma-related appraisals.”

4. The section on mediators and moderators on Page 12 could be clearer in separating mechanisms (typically studied as mediators) and moderators of treatment and the importance of studying both. In the objectives, these two points of view are clearly presented, but in this section in its present form, the reader may end up with the idea that “potential contraindications” refers to mechanisms.

RESPONSE: Thank you for this suggestion. We have clarified what we are referring to in this sub-section, as follows (new text under-lined):

“While change in negative trauma-related misappraisals has been shown to underpin (i.e. mediate) treatment response for cognitive-behavioural therapies^{31 32} (hence our specific secondary hypothesis regarding this mechanism), other potential mechanisms or moderating factors have received scant or no attention. ~~In particular,~~ Potential contraindicative moderators for cognitive-behavioural therapies for PTSD in youth have received comparatively little attention. Shedding light on these factors would inform decision making

about treatments for youth with PTSD and clarify the essential “ingredients” of successful treatment and barriers to change.”

5. “there are no plans for formal interim efficacy or subgroup analyses” (p. 22) – This could be read to suggest that interim analyses might be carried out informally. This is likely not the intention. If, as expected, data will not be looked or analysed during the trial at all, it would be best to state this more clearly.

RESPONSE: We have removed the word “formal” to remove any potential misunderstanding. Thank you for noticing this confusing phrase.

6. The protocol notes that a full statistical analysis plan will be written separately. However, as there will be many outcomes and many analyses, explicitly noting at least the method that will be used to account for the problem of multiple comparisons in the study protocol itself would be valuable. Considering the increasing emphasis on preregistered analyses, it would also be valuable to briefly state which analyses or hypotheses to be examined later have been preregistered and which will be exploratory by nature.

RESPONSE: Thank you for this point. We have only powered the trial to detect a difference on our primary outcome (CRIES-8 scores) at one time point (i.e. post-treatment, approximately five months post-randomisation). We have made this clearer in our Statistical methods sub-section (p19):

“Our primary analysis will compare CT-PTSD with TAU on CRIES-8 scores at post-treatment.”

Moreover, we have made it clear that our other analyses will only be exploratory (p19):

“Analysis of other secondary outcomes, and of putative moderators and mediators, will be considered exploratory.”

Reviewer: 2

I only have 4 areas that require clarification.

1. The first, is that the title and the primary aim of the study is described as to determine if cognitive therapy is effective in the treatment of PTSD of children and adolescents who experienced multiple trauma events. A reference too complex PTSD under unresolved issues in the Background section (p11, line 47) seems to indicate that the study is trying to address both the treatment of children with multiple trauma exposures with a diagnosis of PTSD and also in the process evaluate the characteristics associated with a complex PTSD diagnosis as a secondary aim, such as "disruptions in emotion regulation, relational capacities and self-concept". There seems to however not be any assessment tool to specifically identify complex PTSD as a diagnosis. References to "additional items for measuring dissociation and complex PTSD" is made. So it seems as if the authors expect to treat both PTSD and complex PTSD by identifying children who experienced multiple stressors who meet the criteria for PTSD. It seems that if one argues for two distinct diagnosis (PTSD and Complex PTSD), as some now do, then one would have to assess the effectiveness of both these conditions in two treatment arms, rather than in a combined treatment arm (PTSD)? Perhaps the authors can clarify their thinking around these related diagnosis and the inclusion criteria only being a PTSD diagnosis. The attached NCTU protocol (2017) also does not have treatment of Complex PTSD as an objective, whereas this manuscript has "CT-PTSD

effective in the treatment of complex PTSD" as an objective. This is understandable as the NCTU protocol is dated 2017. This manuscript as an updated version would need to now accommodate the changes around Complex PTSD as a ICD-11 diagnosis (2019) more clearly or at least place this change more clearly in context in terms of the objectives of this study.

RESPONSE: Thank you for this point. Our primary outcome is CRIES-8 scores at post-treatment (approximately five months post-randomisation), and we have only powered the trial to address this question. Our inclusion criteria focus on PTSD, and not complex PTSD. To have focused exclusively on recruiting participants with complex PTSD would have been problematic given the profound uncertainties around optimal methods and tools for its assessment, our impoverished understanding of its importance, and the relatively sparse literature concerning its prevalence when we were designing this trial. All we had to work with, when designing the trial, was a preliminary view of which symptoms the ICD-11 complex PTSD diagnosis would likely contain (e.g. Cloitre et al., 2014, European Journal of Psychotraumatology, <http://dx.doi.org/10.3402/ejpt.v5.25097>). PTSD is an important condition in its own right, and focusing exclusively on the subset of youth with complex PTSD may have made the trial unfeasible.

While we think that reporting outcomes relating to the (relatively new) Complex PTSD construct is important, we should stress that this is only a secondary outcome, and any analyses involving Complex PTSD would be exploratory (see pages 9-10 for primary and secondary outcomes). The extent to which psychological therapies for PTSD, such as CT-PTSD, are also efficacious for complex PTSD is an open question. We would argue that this is an important contribution to the existing literature, given the uncertainties around it at the present time. Our 2017 protocol did include complex PTSD symptoms as an outcome (p9 in the protocol); under objectives (section 5.2.2) we simply referred to "the effect of CT-PTSD on other mental health outcomes and functioning" to cover all other secondary outcomes, which would include complex PTSD symptoms as well depression symptoms, anxiety symptoms etc (i.e. all measures listed on p9 of the protocol, and described more fully on p27).

The items we developed we have shared with other groups, and a recent study of PTSD complex PTSD in looked after children (Hiller et al., 2021, <https://doi.org/10.1111/jcpp.13232>) made use of the questionnaire format of these items for their study, and reported them in their supplementary material. Thus, the items are in the public domain. The items we used to assess dissociation (i.e. depersonalisation and derealisation) are also in the public domain, as they were developed in a previous study (Meiser-Stedman et al., 2019, <https://doi.org/10.1111/jcpp.13054>).

We anticipate using the complex PTSD items we have added to our assessments to report how participants would meet threshold for the complex PTSD diagnosis at baseline, as we believe that this would be informative, i.e. readers may like to know how common this diagnosis is in youth exposed to multiple trauma who meet DSM-5 criteria for PTSD. Moreover, we would believe it would be interesting to explore whether the presence of complex PTSD moderates treatment responsiveness; moderation analysis with respect to complex PTSD will be included in the statistical analysis plan (SAP).

The treatment under evaluation here (CT-PTSD) pays particular attention to trauma-related beliefs. As such, it may be therefore have scope for addressing the negative self-concept symptom that is a constituent symptom of complex PTSD. The extent to which complex PTSD may need an entirely novel treatment is unknown, and the exploratory analyses we will undertake will inform how this question is addressed in the future.

We have not added any new text to the manuscript outlining these points, as complex PTSD is only one secondary outcome of several. However, we have provided a brief outline for the reader on p14 as to the content of the symptoms unique to complex PTSD:

“(that is, items addressing disruptions in emotion regulation, relational capacities and self-concept, that are unique to complex PTSD)”

2. Secondly, a bit more detail on the planned analysis of secondary measures and mediator and moderator analysis would be welcomed.

RESPONSE: We have added the following text on p19:

“Mediation and moderation analyses will be undertaken following the procedures outlined by Kraemer and colleagues.⁶² As with all our secondary analyses, these will be considered exploratory.”

3. Thirdly, as the journal require a start date and proposed completion date of active treatment, perhaps the authors can add this to the manuscript.

RESPONSE: Thank for this point, we have added the following information at the beginning of the manuscript:

“Trial recruitment commenced on the 1st February 2017. It is anticipated that recruitment will continue until June 2021, with 11 month assessments being concluded in May 2022.”

We have also provided some clarification around the timing of the publication of data. Essentially, we did not believe it would be appropriate to delay publication of the post-treatment and 11 month follow up data until we had completed the 29 month assessment (see p25):

“Outcomes from the post-treatment and 11 month post-randomisation assessments may be analysed and published prior to the completion of the 29 month assessments.

4. Fourthly, the authors make no direct reference to the most studied TF-CBT intervention in children and adolescents (Cohen, Mannarino and Deblinger) of the same name. An article published in 2012 in Child Abuse and Neglect by Cohen et al. already describe the relevance of this treatment in the management of complex trauma. More recent RCT's involving TF-CBT have mostly included children with multiple traumas (average of 3.4); (Mannarino quoted in a presentation at NCTSN). To date 21 RCT's of this intervention have been completed in multiple settings ranging from LMICS to EU settings (Netherlands, Norway, Germany). Therefore the motivation behind investigating this particular CT-PTSD treatment manual can be motivated in more detail and perhaps a reference to the results of the pilot study that preceded this study could help motivate this choice?

RESPONSE: Thank you for raising this important issue. Given word constraints we could not justify every decision we took, but we are pleased to explain the choice of CT-PTSD. How much substantive difference there exists between the different TF-CBT packages (e.g. the Cohen/Mannarino/Deblinger manual; Prolonged Exposure for Adolescents; Cognitive Processing Therapy; CT-PTSD) is an open question, particularly given they all place some emphasis on dropping avoidance, some degree of exposure, and some kind of cognitive restructuring. However, there were several reasons for opting to use CT-PTSD. We previously found that CT-PTSD was very efficacious in our (albeit waiting list controlled) RCTs addressing single incident PTSD. Our unpublished pilot study of CT-PTSD for PTSD in youth following multiple traumas suggested that the intervention was acceptable to children and adolescents. Moreover, the intervention appeared to successfully address the primary target for mechanism of interest, trauma-related negative appraisals. This, combined with several published studies that have found that changes in trauma-related appraisals in therapy is a mediator of

treatment response (e.g. Jensen et al., 2018, McLean et al. 2015, Meiser-Stedman et al., 2017, Pfeiffer et al., Smith et al., 2007), suggested that this intervention may be a suitably focused and efficient treatment for PTSD in this population. We also felt that “front-line” clinicians, who may be uncomfortable with “manualised interventions” (Becker et al., 2004; doi:10.1016/S0005-7967(03)00138-4), would appreciate the formulation-based approach of CT-PTSD, i.e. they have the ability to tailor session content to their client’s individual presentation.

We have added some additional text to p8 as follows:

“We felt that CT-PTSD may translate well to “front-line” clinical settings as it employs a formulation-based approach, i.e. clinicians are able to tailor session content (i.e. particular techniques) to their client’s individual presentation which may be particularly helpful when working with clients with more complex histories.”

Reviewer: 3

The authors have registered the study before entry of the first participant. However, I was wondering why the study protocol was not published earlier (recruitment is anticipated to end soon) as a detailed publication of study protocols can reduce publication bias and improve the reproducibility of research.

RESPONSE: Having registered the trial on ISRCTN in 2016, we then focused on recruitment and engaging clinical teams to support recruitment. We then encountered more difficulties with engaging clinical teams and recruiting therapists. Addressing issues around team engagement, therapist retention and recruitment has taken up a considerable amount of time (in particular the addition of further NHS Trusts as recruitment sites). As a result, we have been delayed in getting this protocol submitted. However, as noted above in response to Reviewer 1, we have made only minor changes to our procedures (i.e. the full trial protocol).

I have only minor comments/questions:

1. How were therapists selected? Were therapists randomized to either be trained in CT-PTSD or not? This may influence the comparison as therapists open the to be trained in CT-PTSD might constitute a “special” group of NHS therapists especially interested in EBTs for PTSD in children and adolescents.

RESPONSE: CT-PTSD therapists have not been not randomly selected. We have allowed individual teams to nominate staff to complete the CT-PTSD training. Randomisation of staff is not feasible as teams have to be satisfied they can offer a good level of care in the TAU arm of the trial; randomising therapists may result in a situation where they would end up with TAU clinicians being unequipped to take on a case (as they did not have any training to equip them for work with PTSD). Moreover, therapist randomisation this may further jeopardised recruitment, as we would need to double the number of therapists willing to undergo the therapy training.

We have added the following sentence clarifying this on p12:

“Individual teams will nominate staff members to complete the CT-PTSD training.”

2. How was the stratification value (17-28, 29-40) of the CRIES-8 determined?

RESPONSE: We undertook a pilot study of CT-PTSD for 8-17 year olds with PTSD following multiple trauma; This was before DECRYPT was funded. All nine participants in this pilot study completed the CRIES-8; their mean score at baseline was 28.75. Our stratification cut-off was derived from this figure. We added the following text to the manuscript to reflect this, p17:

“This cut-off for stratification was based on data from an unpublished pilot study that preceded DECRYPT, where the mean score for study participants was 28.75.”

3. The authors could reflect a little more on why they have chosen the CRIES-8 as a primary outcome over the CPSS-I-5 severity score.

RESPONSE: Please see our response to Reviewer 1, Point 2, above.

VERSION 2 – REVIEW

REVIEWER	Samuli Kangaslampi Tampere University, Faculty of Social Sciences / Psychology
REVIEW RETURNED	03-May-2021

GENERAL COMMENTS	The authors have addressed all concerns and suggestions I and other reviewers presented. The amendments and changes made to the manuscript have improved it further and it now describes the study very appropriately and thoroughly. I have no further comments.
---

REVIEWER	J Rossouw Stellenbosch University Faculty of Medicine and Health Sciences, Psychiatry
REVIEW RETURNED	16-May-2021

GENERAL COMMENTS	Thank you for the detailed answers to my enquiries more than satisfied with the reply. I have no further comments other than to wish you well with the continuation of your important study. I look forward to read the results.
--